

# Optimization of polysaccharide conditions and analysis of antioxidant capacity in the co-culture of *Sanghuangporus vaninii* and *Pleurotus sapidus*

Yuantian Lu[1] and Di Liu[1,2]

[1] Agricultural College, Yanbian University, Yanji, Jilin, China
[2] Institute of Edible and Medicinal Fungi, Agricultural College, Yanbian University, Yanji, Jilin, China

## ABSTRACT

Fungal polysaccharides are commonly utilized in the food industry and biomedical fields as a natural and safe immune modulator. Co-culturing is a valuable method for enhancing the production of secondary metabolites. This study used intracellular polysaccharide (IPS) content as a screening index, co-culturing seven different fungi with *Sanghuangporus vaninii*. The seed pre-culture liquid culture time was selected through screening, and conditions were assessed using single factor experimentation, a Plackett-Burman (PB) design, and response surface methodology (RSM) optimization. RSM optimization was conducted, leading to the measurement of antioxidant capacity. Results indicated that the co-culture of *S. vaninii* and *Pleurotus sapidus* exhibited the most effective outcome. Specifically, pre-culturing *S. vaninii* and *P. sapidus* seed cultures for 2 days and 0 days, respectively, followed by co-culturing, significantly increased IPS content compared to single-strain culturing. Further optimization of co-culture conditions revealed that yeast extract concentration, liquid volume, and *S. vaninii* inoculum ratio notably influenced IPS content in the order of yeast extract concentration > liquid volume > *S. vaninii* inoculum ratio. Under the optimal conditions, IPS content reached 69.9626 mg/g, a 17.04% increase from pre-optimization co-culture conditions. Antioxidant capacity testing demonstrated that co-cultured IPS exhibited greater scavenging abilities for DPPH and ABTS free radicals compared to single strain cultures. These findings highlight the potential of co-culturing *S. vaninii* and *P. sapidus* to enhance IPS content and improve antioxidant capacity, presenting an effective strategy for increasing fungal polysaccharide production.

# INTRODUCTION

*Sanghuangporus* spp. are a type of large edible and medicinal fungi that belong to the family Hymenochaetaceae and the genus *Sanghuangporus* (*Zhou et al., 2016*). The medicinal properties of this fungus were first documented in traditional Chinese medicine texts, such as 'Shen Nong's Materia Medica,' and are internationally recognized for their

Corresponding author
Di Liu, liudi@ybu.edu.cn

potent anti-cancer effects (*Wu et al., 2012*, *2022*). Sanghuang contains a range of bioactive compounds, including polysaccharides (*Yang et al., 2020*), flavonoids (*Wu et al., 2019*), polyphenols (*Liu et al., 2017*), terpenoids (*Rajachan et al., 2020*), steroidal compounds (*Thanh et al., 2018*), and nitrogen-containing compounds (*Li et al., 2015*). Among these, polysaccharides are one of the key bioactive components responsible for the pharmacological effects of Sanghuang (*Wang et al., 2022*). This fungus exhibits various pharmacological properties, such as anti-tumor effects (*Chakraborty et al., 2019*), anti-inflammatory properties (*Hou et al., 2020*), hepatoprotective effects (*Yuan et al., 2019*), immune-modulating properties (*Meng, Liang & Luo, 2016*), and antioxidant activity (*Cai et al., 2019*). The efficacy of this fungus has garnered significant interest in both pharmaceutical research and the health product industry, both domestically and internationally. *Sanghuangporus vaninii* (*S. vaninii*) has become one of the most sought-after varieties of this fungus in the market due to its ease of cultivation and strong biological activity.

*Pleurotus sapidus* (*P. sapidus*) is an edible species that is classified under the genus *Pleurotus* and the family Pleurotaceae (*Song et al., 2017*). The *Pleurotus* genus demonstrates a wide range of species diversity, with approximately 773 recorded entries as of December 17, 2023 (source: http://www.indexfungorum.org/Names/Names.asp). However, only about 50 of these species are widely accepted (*Guzmán, 2000*). *Pleurotus* species are rich in nutrients such as polysaccharides, proteins, amino acids, and fatty acids. These components play a crucial role in enhancing immune function, delaying aging, combating fatigue, exhibiting anti-tumor activity, regulating blood sugar and lipid levels, and contributing to chemoprevention (*Xu et al., 2014*; *Im et al., 2014*; *Lin et al., 2014*).

Microbial co-culturing involves culturing two or more strains (such as fungus-fungus, fungus-bacteria, and bacteria-bacteria) in the same vessel (*Mantravadi et al., 2019*), which is known to enhance the production of secondary metabolites (*Bertrand et al., 2014*; *Caudal, Tapissier-Bontemps & Edrada-Ebel, 2022*). *Guo et al. (2021)* demonstrated that co-culturing *Sanghuangporus lonicericola* and *Cordyceps militaris* resulted in significantly higher mycelial polysaccharide yields compared to monocultures, with increases of 69.95% and 59.63%, respectively. *Wu et al. (2021)* observed that co-culturing *Ganoderma lucidum* and *Flammulina velutipes* led to reduced biomass production but increased extracellular polysaccharide (EPS) production compared to monocultures. Therefore, microbial co-culturing is an effective strategy for enhancing polysaccharide production in fungi.

Response surface methodology (RSM) is a valuable tool for exploring the relationship between different variables and for optimizing culture medium components and co-culture conditions. Successful cases have demonstrated increased production of secondary metabolites in microbial co-cultures using RSM strategies. For instance, *Pan et al. (2023)* used RSM to develop a statistical model for the co-culture conditions of *Streptomyces albulus* IFO 14147 and *Corynebacterium glutamicum* CICC 10064, resulting in a 31.47% higher yield of ε-poly-L-lysine at 27.07 ± 0.47 g/L compared to the single-strain yield. Similarly, *Li et al. (2020)* optimized the co-culture conditions of *Trichoderma atroviride* SG3403 and *Bacillus subtilis* 22 using RSM, achieving a 54.22% inhibition rate against *Fusarium graminearum* in the optimized co-culture fermentation

broth, a 9% improvement over the original conditions. These findings highlight the ability of RSM to induce and enhance the production of active secondary metabolites in microbial co-cultures.

In this study, a strain of *P. sapidus* was screened and obtained for co-cultivation with *S. vaninii* to enhance intracellular polysaccharide (IPS) content. A Seed pre-culture liquid culture time of *S. vaninii* and *P. sapidus* was also screened. Single-factor experiments were conducted to optimize concentrations of glucose, yeast extract powder, $MgSO_4 \cdot 7H_2O$, $KH_2PO_4$, as well as initial pH, inoculum volume, proportion of *S. vaninii* inoculum volume, and liquid volume. Key factors were identified through a Plackett-Burman (PB) design, followed by RSM to determine the antioxidant capacity of IPS in the co-culture and monoculture under optimal conditions. This research presents a novel culture strategy for enhancing fungal polysaccharide production and paves the way for the advancement of technologies related to fungal co-culturing and polysaccharide production.

## MATERIALS AND METHODS

### Test strains

*Sanghuangporus vaninii* (ST), *Pleurotus sapidus* (BLZ), *Pleurotus ostreatus* (P8), *Flammulina filiformis* (BJZ), *Irpex lacteus* (BPCJ), *Hericium erinaceus* (HT3), *Lentinus edodes* (XG), and *Panellus edulis* (DM) were obtained from the Institute of Edible and Medicinal Fungi, Agricultural college, Yanbian University, Yanji, Jilin, China. Each strain was identified using a specific code.

### Culture media

The agar plate culture medium used in this study consisted of 200 g/L potato, 20 g/L glucose, and 20 g/L agar.

The liquid culture medium contained 5 g/L yeast extract, 20 g/L glucose, 1.5 g/L $KH_2PO_4$, and 0.5 g/L $MgSO_4 \cdot 7H_2O$.

### Activation of strains

Following removal from the 4 °C refrigerator, the test strains were allowed to equilibrate at room temperature for 2 h to restore their vitality. Subsequently, clumps measuring 0.5 cm × 0.5 cm were aseptically taken using an inoculation spatula and transferred to a fresh plate medium. The petri dishes were then placed in a smart artificial incubator set at 28 °C for incubation.

### Seed culture fluid

Under aseptic conditions, a 9 mm hole was punched to extract one activated fungal mycelial block of *I. lacteus*, *P. sapidus*, and *P. ostreatus*, as well as five fungal mycelial blocks of *S. vaninii*, *H. erinaceus*, *F. filiformis*, *L. edodes*, and *P. edulis*, which were then transferred into the liquid filling medium. The liquid culture medium, with a volume of 100 mL, was placed in a constant temperature shaking incubator set at 28 °C and 150 rpm for cultivation. *Irpex lacteus* was cultured for 4 days, while *P. sapidus* and *P. ostreatus* were cultured for 5 days, and *S. vaninii*, *H. erinaceus*, *F. filiformis*, *L. edodes*, and *P. edulis* were

cultured for 7 days. Once the culture was complete, it was homogenized into a suspension using an adjustable high-speed homogenizer to create a seed liquid.

## Liquid co-culture

The seed liquid was transferred to a liquid medium with a volume of 80 mL, based on an inoculum volume of 20% (v/v), to create the seed pre-culture liquid. Each strain was then individually cultured in a liquid medium with a filling volume of 80 mL using an inoculum volume of 20% (v/v). *Sanghuangporus vaninii* was used as a consistent strain. Subsequently, *S. vaninii* and other strains were each inoculated, with an inoculation volume of 10% (v/v), from the seed pre-culture liquid and transferred to a liquid medium with a volume of 80 mL for co-culturing. The shake flasks were then placed in a constant temperature shaking incubator set at 28 °C and 150 rpm for a duration of 7 days.

## Seed pre-culture liquid culture times screening

As shown in Table 1, liquid co-culture experiments were conducted with *S. vaninii* and *P. sapidus*. Combinations 1–10 involved a *S. vaninii* and *P. sapidus* co-culture, combination 11 was a *P. sapidus* monoculture, and combination 12 was a *S. vaninii* monoculture. The subsequent operational procedures remained consistent. All shake flasks were incubated in a constant temperature shaking incubator at 28 °C and 150 rpm for a duration of 7 days.

## Single-factor experiment

This experiment investigated the effect of eight factors on IPS content in the co-culture. The fundamental culture conditions included glucose concentration (20 g/L), yeast extract concentration (5 g/L), $MgSO_4 \cdot 7H_2O$ concentration (0.5 g/L), $KH_2PO_4$ concentration (1.5 g/L), initial pH (natural), inoculation volume (v/v, 20%), *S. vaninii* inoculum ratio (50%), and liquid loading volume (100 mL).

The levels of each factor in the single-factor experiment were as follows: glucose concentration (10, 20, 30, 40, 50, 60, 70 g/L); yeast extract concentration (2, 5, 8, 11, 14, 17, 20 g/L); $MgSO_4 \cdot 7H_2O$ concentration (0.2, 0.5, 1, 1.5, 2, 2.5, 3 g/L); $KH_2PO_4$ concentration (0.5, 1, 1.5, 2, 2.5, 3, 3.5 g/L); initial pH (3, 4, 5, 6, 7, 8, 9, 10, 11); inoculation volume (v/v, 5, 10, 15, 20, 25, 30, 35 mL); *S. vaninii* inoculum ratio (20%, 30%, 40%, 50%, 60%, 70%, 80%); liquid loading volume (50, 75, 100, 125, 150, 175, 200 mL).

## PB design

The PB design identified the essential factors that impacted the IPS content. These factors are represented as $X_1-X_8$ in Table 2. Each independent variable was assigned two levels, high (+) and low (−), while the IPS content was the responsive variable.

## RSM optimization design

Based on the results of the single factor experiment and PB design, an RSM experiment was conducted. The independent variables in this experiment were yeast extract concentration, liquid loading volume, and *S. vaninii* inoculum ratio, while the responsive variable was IPS content. The levels of each factor are shown in Table 3.

**Table 1 ST and BLZ seed pre-culture liquid culture time.**

| Combination | ST seed pre-culture liquid culture time (d) | BLZ seed pre-culture liquid culture time (d) |
|---|---|---|
| 1 | 0 | 0 |
| 2 | 1 | 1 |
| 3 | 2 | 2 |
| 4 | 3 | 3 |
| 5 | 1 | 0 |
| 6 | 2 | 1 |
| 7 | 3 | 2 |
| 8 | 2 | 0 |
| 9 | 3 | 1 |
| 10 | 3 | 0 |
| 11 | – | 7 |
| 12 | 7 | – |

Note:
Combinations 1–10 are ST and BLZ co-culture, combination 11 is BLZ single culture, and combination 12 is ST single culture.

**Table 2 Factors and levels of the Placket-Burman experimental design.**

| Variables | | | Levels | |
|---|---|---|---|---|
| Factor | Name | Unit | −1 | +1 |
| $X_1$ | Glucose concentration | g/L | 24 | 30 |
| $X_2$ | Yeast extract concentration | g/L | 6.4 | 8 |
| $X_3$ | $KH_2PO_4$ concentration | g/L | 0.8 | 1 |
| $X_4$ | $MgSO_4 \cdot 7H_2O$ concentration | g/L | 1.6 | 2 |
| $X_5$ | Initial pH | – | 7.2 | 9 |
| $X_6$ | Liquid loading volume | mL | 80 | 100 |
| $X_7$ | Inoculation volume | % | 8 | 10 |
| $X_8$ | ST inoculum ratio | % | 56 | 70 |

**Table 3 Factors and levels of the Box-Behnken experimental design.**

| Variables | | | Levels | | |
|---|---|---|---|---|---|
| Factor | Name | Unit | −1 | 0 | +1 |
| A | Yeast extract concentration | g/L | 5 | 8 | 11 |
| B | ST inoculum ratio | % | 60 | 70 | 80 |
| C | Liquid loading volume | mL | 75 | 100 | 125 |

## IPS content determination

IPS content was determined using the methods outlined by *Wang (2021)*, with slight modifications. The cultured fermentation broth was filtered using four layers of gauze. The mycelial balls were then rinsed with distilled water and dried in a drying oven at 60 °C

until a constant weight was achieved. After cooling, the mycelium balls were weighed to obtain their biomass. Subsequently, the dried mycelium balls were crushed and ground through a 60-mesh standard sieve, and samples were collected for the determination of IPS content.

The hot water extraction method was used in this experiment for IPS extraction. A sample weighing 0.3 g was accurately measured and placed in a centrifuge tube. Distilled water was added to achieve a material-to-liquid ratio of 1:30 (g/mL). The centrifuge tube was then placed in a 90 °C water bath for a 2-h extraction period. Afterward, the mixture was centrifuged at 8,000 r/min for 30 min to collect the supernatant. This extraction process was repeated twice, and the supernatants were combined and concentrated in an electrothermal constant temperature drying oven to one-fifth of the original volume. Next, four times the volume of absolute ethanol was added, and the mixture was allowed to stand at 4 °C for 12 h. The mixture was then centrifuged at 8,000 r/min for 30 min to collect the precipitate, which was then placed in a drying box to completely evaporate the ethanol. Distilled water was added to redissolve the precipitate, resulting in the IPS test solution.

For the glucose standard solution (0.1 mg/mL), 0.1 g of glucose standard was weighed and then placed into a 100 mL beaker. Distilled water was then added to dissolve and dilute the solution to 1,000 mL and then stored at 4 °C. Next, 0, 0.2, 0.4, 0.6, 0.8, and 1 mL of the glucose standard solution were pipetted into separate test tubes. Distilled water was added to each test tube to reach a total volume of 1 mL and then each test tube was shaken well. After that, 1 mL of 5% phenol and 5 mL of concentrated sulfuric acid were added to each test tube in sequence. After being mixed well, the test tubes were bathed in boiling water for 15 min. Finally, the test tubes were cooled quickly under running water and then sat at room temperature for 10 min. The absorbance value was measured at a wavelength of 490 nm using distilled water as a blank. The glucose standard curve was plotted using the glucose content (mg) as the abscissa and the absorbance value as the ordinate. The linear regression equation of the glucose standard was $Y = 10.185X - 0.0453$ ($R^2 = 0.9927$). The IPS content was calculated according to the following formula:

$$\text{IPS content (mg/g)} = \frac{m_1 \times V_1}{m \times V} \times N$$

In the formula: $m_1$—glucose content calculated from the standard curve, mg; $V_1$—constant volume, mL; $N$—dilution factor; $m$—sample mass, g; $V$—volume of sample solution taken during measurement, mL.

## Measurement of antioxidant capacity

### Measurement of DPPH free radical scavenging capacity

For measuring DPPH free radical scavenging capacity, the methods described by *Zeng et al. (2012)* were followed, with slight adjustments. Samples were accurately weighed and diluted with distilled water to create solutions of varying concentrations. DPPH was weighed precisely and mixed with absolute ethanol to create a 0.1 mmol/L DPPH solution. Subsequently, 2 mL of a sample solution at a specific concentration was mixed with 2 mL of the DPPH solution, thoroughly mixed, and incubated in the dark at 30 °C for 30 min.

The absorbance at 517 nm, denoted as $A_1$, was then measured. The absorbance when distilled water was used instead of the DPPH solution was recorded as $A_2$, and the absorbance when distilled water was used in place of the sample solution was recorded as $A_0$. The DPPH free radical scavenging rate was calculated using the following formula:

$$\text{DPPH free radical scavenging (\%)} = \left(1 - \frac{A_1 - A_2}{A_0}\right) \times 100$$

### Measurement of ABTS free radical scavenging capacity

For measuring ABTS free radical scavenging capacity, the methods of *Jo, Cho & Chun (2021)* were followed, with slight modifications. Equal amounts of 7 mmol/L ABTS solution and 2.45 mmol/L $K_2O_8S_2$ solution were mixed evenly and left to stand in the dark at room temperature for 12 h to obtain the ABTS mother solution. The ABTS mother solution was then diluted with distilled water until its absorbance at 734 nm was 0.7 ± 0.02, recorded as $A_0$, to obtain the ABTS working solution. Sample solutions of different concentrations (1 mL each) were mixed evenly with 4 mL of ABTS working solution, placed in the dark at 30 °C for 6 min, and the absorbance at 734 nm was measured and recorded as $A$. The ABTS free radical scavenging rate was calculated using the following formula:

$$\text{ABTS free radical scavenging (\%)} = \left(1 - \frac{A}{A_0}\right) \times 100$$

## Statistical analysis

Data enumeration was performed using Excel, and single factor variance analysis (ANOVA) was conducted using SPSS 26.0 software. PB and RSM experiments were conducted using Expert 13 software. $^*p < 0.05$ and $^{**}p < 0.01$ indicated significant and highly significant differences, respectively. Each experiment was replicated three times.

# RESULTS

## Enhancement of IPS content by co-cultivation of *S. vaninii* and *P. sapidus*

In this study, seven different strains were chosen to be co-cultured with *S. vaninii*. Strain *S. vaninii* was co-cultured with the seven candidate strains, resulting in the inhibition of *S. vaninii* growth, while all candidate strains exhibited normal growth (Fig. 1). As shown in Table 4, when *S. vaninii* was co-cultured with the candidate strains, only the IPS content of *I. lacteu*, *P. sapidus*, and *H. erinaceus* co-cultures was higher than the IPS content of the individual strains. The IPS content of the *S. vaninii* and *P. sapidus* co-culture was 370.05% higher than that of *S. vaninii* alone and 11.4% higher than *P. sapidus* alone. Additionally, the biomass of the *S. vaninii* and *P. sapidus* co-culture was 59.57% higher than that of *S. vaninii* alone and 5.67% higher than *P. sapidus* alone. Based on these results, *S. vaninii* and *P. sapidus* were chosen to enhance the IPS content of further co-culture experiments.

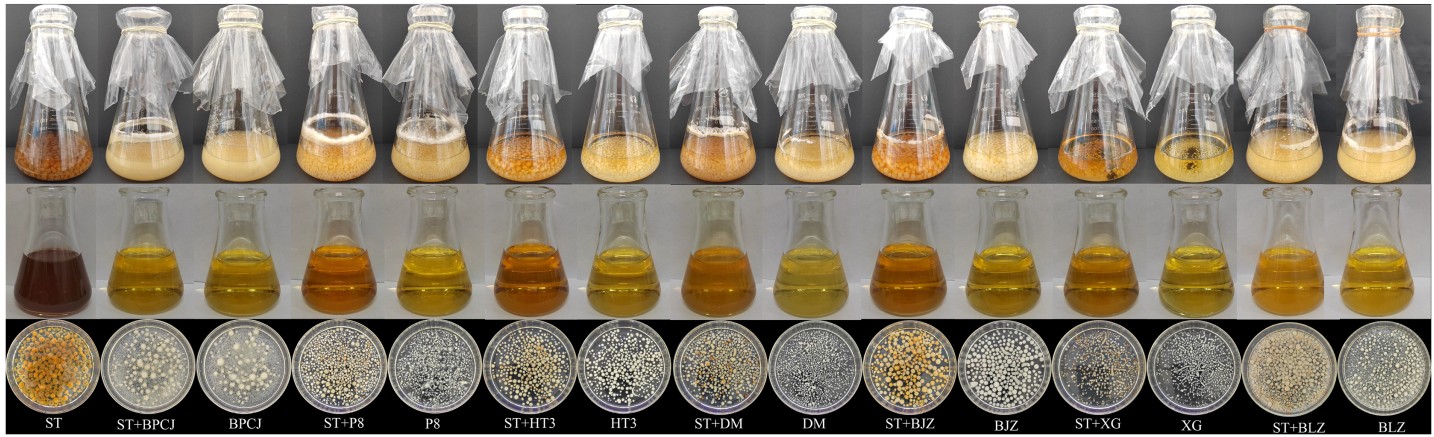

**Figure 1** ST was purely cultured and co-cultured with each of the strains to be screened in liquid.

**Table 4 Biomass and IPS content of ST in pure culture and co-culture with each strain to be screened.**

| No. | Test strains | Biomass (g/100 mL) | IPS content (mg/g) |
|-----|--------------|--------------------|--------------------|
| 1 | ST | 0.5775 ± 0.0150ef | 12.7286 ± 0.3774k |
| 2 | BPCJ+ST | 0.5377 ± 0.0165g | 21.3047 ± 0.6351h |
| 3 | BPCJ | 0.5640 ± 0.0112fg | 20.1241 ± 0.6093i |
| 4 | P8+ST | 0.8174 ± 0.0251c | 23.1432 ± 0.4273fg |
| 5 | P8 | 0.8174 ± 0.0176c | 25.1078 ± 0.6944d |
| 6 | HT3+ST | 0.4223 ± 0.0093h | 24.5098 ± 0.1535de |
| 7 | HT3 | 0.3406 ± 0.0102i | 23.7068 ± 0.5603ef |
| 8 | DM+ST | 0.6848 ± 0.0171d | 13.3412 ± 0.3346k |
| 9 | DM | 0.5987 ± 0.0148e | 15.7817 ± 0.2976j |
| 10 | BJZ+ST | 0.4367 ± 0.0083h | 16.3433 ± 0.4364j |
| 11 | BJZ | 0.7005 ± 0.0238d | 22.7538 ± 0.5149g |
| 12 | XG+ST | 0.0420 ± 0.0051j | 20.1931 ± 0.3185i |
| 13 | XG | 0.0503 ± 0.0063j | 27.1388 ± 0.6252c |
| 14 | BLZ+ST | 0.9215 ± 0.0322a | 59.8302 ± 0.8376a |
| 15 | BLZ | 0.8729 ± 0.0181b | 53.7090 ± 0.8054b |

**Note:**
Different letters indicate significant differences in means ($p < 0.05$).

## Screening of *S. vaninii* and *P. sapidus* seed pre-cultures for culture time

Different fungi exhibit varying growth rates, and the efficacy of fungal co-cultures are greatly influenced by the timing of inoculation. To ensure a thorough co-culturing of fungi with different growth rates, the inoculation epochs of *S. vaninii* and *P. sapidus* were screened.

There were 10 co-culture combinations of *S. vaninii* and *P. sapidus* (combinations 1–10). When the seed pre-culture solution intervals of *S. vaninii* and *P. sapidus* were the same, the color of the fermentation filtrate gradually darkened and the size of the bacterial

 

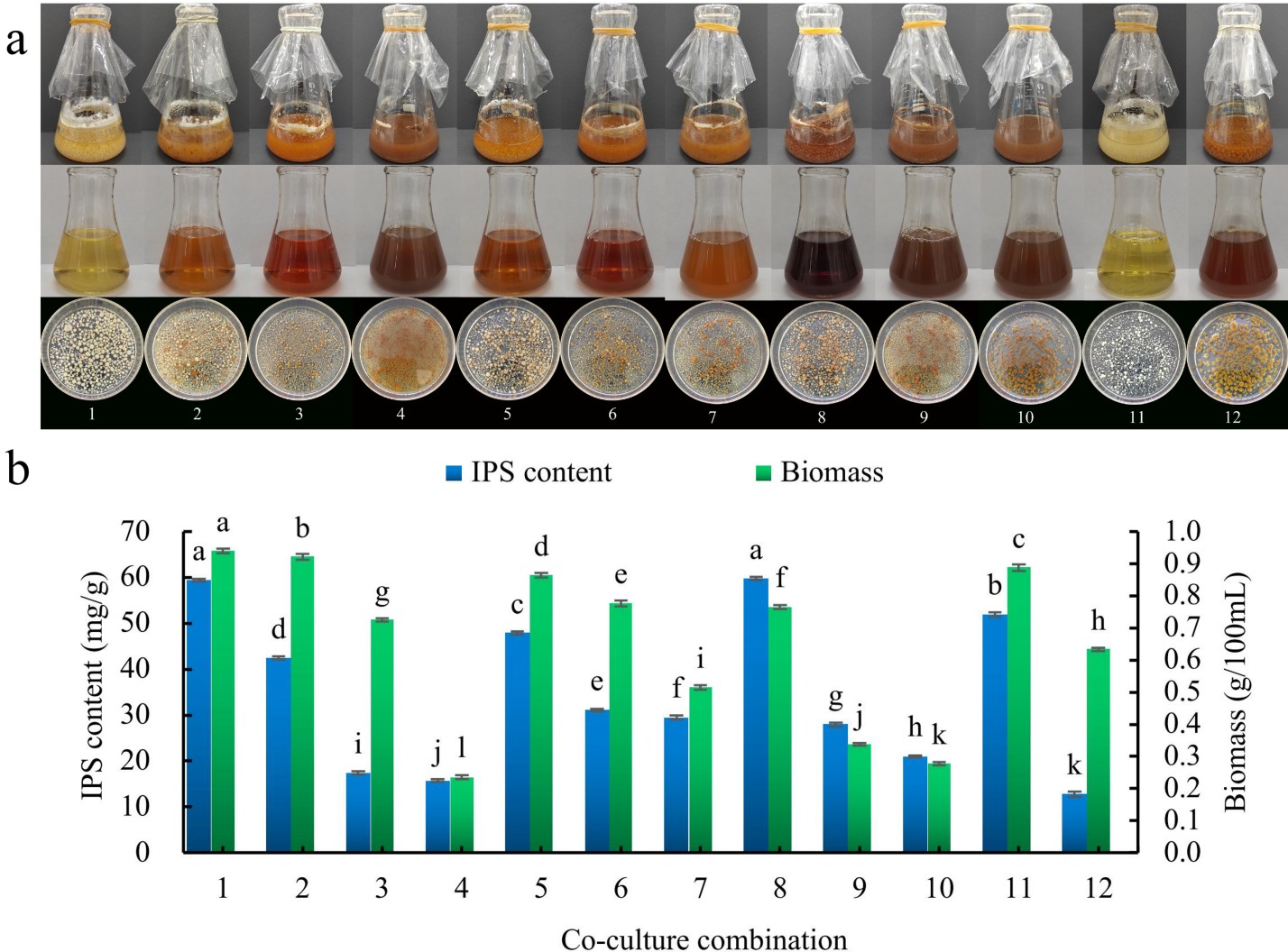

**Figure 2 Seed pre-culture liquid culture times screening.** (A) Liquid culture of different co-culture combinations; (B) effect of different co-culture combinations on IPS content and biomass. Different lowercase letters indicate significant differences in means ($p < 0.05$). Combinations 1–10 are *S. vaninii* and *P. sapidus* co-culture, combination 11 is *P. sapidus* pure culture, and combination 12 is *S. vaninii* pure culture.

balls decreased as culture time increased, showing a more pronounced inhibitory effect. The color of *S. vaninii* bacterial balls shifted from yellow to orange-red, while the color of *P. sapidus* bacterial balls changed from white to yellow or yellow brown (Fig. 2A). Additionally, with longer seed pre-culture medium culture times, both the biomass and IPS content exhibited a decreasing trend (Fig. 2B). Combination 1 showed significantly higher biomass compared to combination 8, although there was no significant difference in the IPS content between the two combinations. The co-culture effect was more prominent in combination 8 than in combination 1. Consequently, combination 8 was chosen as the test combination, where the seed pre-culture fluids of *S. vaninii* and *P. sapidus* were cultured for 2 days and 0 days, respectively, before co-culturing. In combination 8, the IPS content was 4.68 times higher than that of the *S. vaninii* culture alone (combination 12)

and 1.15 times higher than that of the *P. sapidus* culture alone (combination 11), while the biomass was 1.21 times higher than that of the *S. vaninii* culture alone and 0.86 times higher than that of the *P. sapidus* culture alone.

## Single-factor experiments

### Glucose concentration screening

According to the results (Fig. 3A), the IPS content showed a gradual increase within the glucose concentration range of 10–30 g/L. At a glucose concentration of 30 g/L, the IPS content reached its highest value of 57.9784 ± 0.4536 mg/g. However, within the 30–70 g/L glucose concentration range, the IPS content decreased. Conversely, the biomass consistently increased from 0.6604 ± 0.0100 g/100 mL to 1.0310 ± 0.0455 g/100 mL within the 10–70 g/L glucose concentration range. Based on the selection criterion of IPS content, the optimal glucose concentration was determined to be 30 g/L.

### Yeast extract concentration screening

The results indicate (Fig. 3B) that both the IPS content and biomass increased as the yeast extract concentration increased in the 2–8 g/L range. At a yeast extract concentration of 8 g/L, both the IPS content and biomass reached their maximum values, measuring 59.4230 ± 0.4821 mg/g and 1.1008 ± 0.0902 g/100 mL, respectively. However, when the yeast extract concentration exceeded 8 g/L and ranged from 8–20 g/L, the IPS content showed a decreasing trend, while the biomass remained relatively stable. Therefore, the optimal yeast extract concentration was determined to be 8 g/L.

### $MgSO_4 \cdot 7H_2O$ concentration screening

The results indicate (Fig. 3C) that there was an increasing trend in IPS content as $MgSO_4 \cdot 7H_2O$ concentration increased in the 0.2–2 g/L range. The IPS content reached its highest value of 62.0891 ± 0.4404 mg/g when the concentration of $MgSO_4 \cdot 7H_2O$ was 2 g/L, after which it decreased. Similarly, the biomass showed an increasing trend as $MgSO_4 \cdot 7H_2O$ concentration increased in the 0.2–2.5 g/L range. The biomass reached its peak value of 0.8379 ± 0.0079 g/100 mL when the concentration of $MgSO_4 \cdot 7H_2O$ was 2.5 g/L and then remained stable. Based on the selection criterion of IPS content, the optimal concentration of $MgSO_4 \cdot 7H_2O$ was determined to be 2 g/L.

### $KH_2PO_4$ concentration screening

As $KH_2PO_4$ concentration increased from 0.5 to 1.5 g/L of $KH_2PO_4$, both IPS content and biomass showed an increasing trend. However, at a $KH_2PO_4$ concentration of 1.5 g/L, both the IPS content and biomass reached their maximum values, measuring 57.6404 ± 0.5523 mg/g and 0.7956 ± 0.0053 g/100 mL, respectively. As $KH_2PO_4$ concentration increased from 1.5 to 3.5 g/L, IPS content decreased, while biomass initially decreased and then increased (Fig. 3D). Therefore, the optimal concentration of $KH_2PO_4$ was determined to be 1.5 g/L.

### Initial pH screening

The concentration of IPS showed an increasing trend across the pH range of 3–9. Initially, the IPS concentration was relatively low within the pH range of 3–4. However, it rapidly

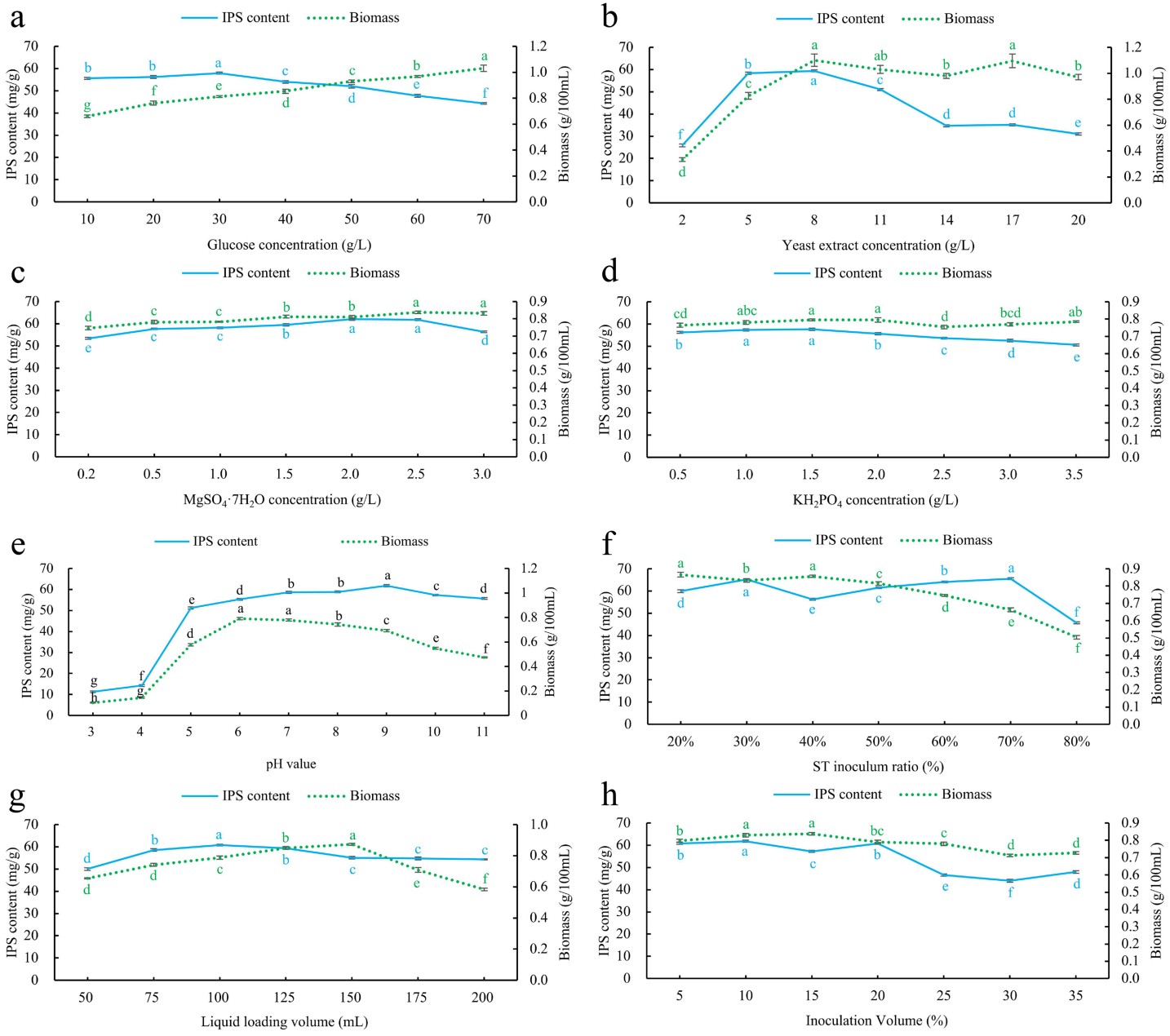

**Figure 3 Effect of single factor index screening on co-culture IPS content and biomass.** (A) Glucose concentration; (B) yeast extract concentration; (C) MgSO₄·7H₂O concentration; (D) KH₂PO₄ concentration; (E) pH; (F) *S. vaninii* inoculum ratio; (G) Liquid loading volume; (H) inoculum volume. Different lowercase letters indicate significant differences in means ($p < 0.05$).

increased as the pH increased from 4 to 5, followed by a gradual increase within the pH range of 5–9. The highest IPS content of 61.7806 ± 0.5062 mg/g was observed at pH 9, after which it started to decrease. Conversely, biomass showed an upward trajectory within the pH range of 3–6, reaching its highest value of 0.7904 ± 0.0104 g/100 mL at pH 6 before starting to decline (Fig. 3E). Based on the evaluation of IPS content, the optimal pH was determined to be nine.

### Sanghuangporus vaninii inoculum ratio screening

Within the 20–80% inoculation ratio range of *S. vaninii*, the IPS content showed an overall ascending-descending-ascending-descending trend. At a 30% and 70% inoculation ratio, the IPS content reached peak values of 65.2716 ± 0.3909 mg/g and 65.5855 ± 0.4270 mg/g, respectively, with no significant difference between these two peaks (Fig. 3F). Conversely, biomass consistently decreased. Due to the faster mycelial growth rate of *P. sapidus* compared to *S. vaninii*, the biomass increased as the proportion of *P. sapidus* inoculation increased. Therefore, to establish a more comprehensive co-culture state of the two strains and maximize their interaction, the optimal inoculation ratio for *S. vaninii* was determined to be 70%.

### Liquid loading volume screening

The IPS content showed an increasing trend as the liquid volume increased in the 50–100 mL range. At a liquid volume of 100 mL, the IPS content reached its peak at 60.7687 ± 0.3657 mg/g, and then decreased as liquid volume increased. Similarly, the biomass increased as the liquid volume increased from 50 to 150 mL, reaching its highest value of 0.8736 ± 0.0064 g/100 mL before declining (Fig. 3G). Based on the evaluation criterion of IPS content, the optimal liquid volume was determined to be 100 mL.

### Inoculum volume screening

IPS content showed an ascending-descending-ascending-descending trend as inoculum volume increased from 5% to 35%. At inoculum volumes of 10% and 20%, the IPS content reached peak values of 61.8494 ± 0.4158 mg/g and 60.8487 ± 0.4427 mg/g, respectively, with a significant difference between these two peaks. Conversely, biomass increased as inoculum volume increased from 5% to 15%, reaching its highest point at 15% with a value of 0.8378 ± 0.008 g/100 mL, after which it started to decline (Fig. 3H). Based on the evaluation criterion of IPS content, the optimal inoculum volume was determined to be 10%.

The optimal culture conditions were determined through Single-factor experiments, with the following parameters: glucose concentration of 30 g/L, yeast extract concentration of 8 g/L, $MgSO_4 \cdot 7H_2O$ concentration of 2 g/L, $KH_2PO_4$ concentration of 1.5 g/L, pH of 9, *S. vaninii* inoculum ratio of 70%, Liquid loading volume 100 mL, and inoculum volume of 10%.

### PB design

In this study, a PB design strategy was used to systematically investigate the effects of different elements on IPS content in the co-culture and identify which factors significantly affected IPS content. The experimental design and interpretation of results were conducted using Design Expert 13 software. The specific methodologies and outcomes of the experiments are presented in Table 5.

A variance analysis was performed on the experimental data, and the results are shown in Table 6. The high values of the model confirmation coefficient $R^2$ and the adjusted coefficient Adj $R^2$ indicate that the model fits well and the results are highly reliable. Among the elements studied, $X_2$, $X_6$, and $X_8$ had significant influences on the response

**Table 5 Placket-Burman experimental design and results.**

| Run. order | X₁ | X₂ | X₃ | X₄ | X₅ | X₆ | X₇ | X₈ | Response value |
|---|---|---|---|---|---|---|---|---|---|
| | Glucose concentration (g/L) | Yeast extract concentration (g/L) | KH₂PO₄ concentration (g/L) | MgSO₄·7H₂O concentration (g/L) | Initial pH | Liquid loading volume (mL) | Inoculation volume (%) | ST inoculum ratio (%) | IPS content (mg/g) |
| 1 | 30 | 8 | 0.8 | 1.6 | 9 | 100 | 8 | 56 | 52.9959 |
| 2 | 24 | 6.4 | 1 | 1.6 | 9 | 100 | 8 | 70 | 44.9510 |
| 3 | 24 | 8 | 0.8 | 2 | 7.2 | 100 | 10 | 70 | 58.0327 |
| 4 | 30 | 6.4 | 0.8 | 1.6 | 7.2 | 100 | 10 | 70 | 64.0629 |
| 5 | 30 | 6.4 | 1 | 2 | 7.2 | 80 | 8 | 70 | 52.0004 |
| 6 | 24 | 6.4 | 0.8 | 1.6 | 7.2 | 80 | 8 | 56 | 58.4561 |
| 7 | 24 | 8 | 1 | 1.6 | 9 | 80 | 10 | 70 | 29.6220 |
| 8 | 30 | 8 | 0.8 | 2 | 9 | 80 | 8 | 70 | 33.7195 |
| 9 | 30 | 6.4 | 1 | 2 | 9 | 100 | 10 | 56 | 64.1685 |
| 10 | 24 | 6.4 | 0.8 | 2 | 9 | 80 | 10 | 56 | 59.6148 |
| 11 | 30 | 8 | 1 | 1.6 | 7.2 | 80 | 10 | 56 | 43.6575 |
| 12 | 24 | 8 | 1 | 2 | 7.2 | 100 | 8 | 56 | 48.8338 |

**Table 6 First-order polynomial model analysis of Placket-Burman test results.**

| Source | | Sum of squares | F-value | p-value |
|---|---|---|---|---|
| Model | | 1,329.81 | 10.17 | 0.0413* |
| X₁ | Glucose concentration | 10.26 | 0.63 | 0.4861 |
| X₂ | Yeast extract concentration | 486.32 | 29.76 | 0.0121* |
| X₃ | KH₂PO₄ concentration | 158.77 | 9.72 | 0.0526 |
| X₄ | MgSO₄·7H₂O concentration | 42.65 | 2.61 | 0.2046 |
| X₅ | Initial pH | 133.14 | 8.15 | 0.0649 |
| X₆ | Liquid loading volume | 261.1 | 15.98 | 0.0281* |
| X₇ | Inoculation volume | 66.28 | 4.06 | 0.1375 |
| X₈ | ST inoculum ratio | 171.3 | 10.48 | 0.0479* |
| R² | | | 0.9644 | |
| Adj R² | | | 0.8696 | |

Note:
*$p < 0.05$.

variable. The extent of their influence was ranked as $X_2 > X_6 > X_8$. Therefore, future experiments will focus on examining the effects of these three elements on the response variable to determine the optimal culture conditions.

## RSM optimization experiment
### RSM experimental design and results
This investigation used the Box-Behnken central composite design strategy with adjustable variables including yeast extract concentration, *S. vaninii* inoculum ratio, and liquid

**Table 7 Response surface methodology test design and results.**

| Run. order | Variables | | | Response value |
|---|---|---|---|---|
| | A: Yeast extract concentration (g/L) | B: ST inoculum ratio (%) | C: Liquid loading volume (mL) | IPS content (mg/g) |
| 1 | 5 | 70 | 125 | 61.6408 |
| 2 | 5 | 80 | 100 | 60.8699 |
| 3 | 11 | 70 | 125 | 34.7527 |
| 4 | 8 | 70 | 100 | 66.7635 |
| 5 | 8 | 60 | 125 | 47.3028 |
| 6 | 11 | 60 | 100 | 50.6033 |
| 7 | 8 | 70 | 100 | 65.7598 |
| 8 | 8 | 80 | 125 | 56.9552 |
| 9 | 8 | 70 | 100 | 65.7612 |
| 10 | 8 | 70 | 100 | 66.5578 |
| 11 | 8 | 70 | 100 | 65.2032 |
| 12 | 11 | 70 | 75 | 32.7527 |
| 13 | 5 | 60 | 100 | 66.1607 |
| 14 | 11 | 80 | 100 | 43.0082 |
| 15 | 8 | 60 | 75 | 51.5302 |
| 16 | 8 | 80 | 75 | 34.3047 |
| 17 | 5 | 70 | 75 | 42.7653 |

loading volume. A three-factor, three-level experimental design was implemented to refine the culture conditions, with IPS content as the response parameter. The results are presented in Table 7. The generated IPS content was evaluated and fitted using Design Expert 13 software.

The regression equation for IPS content ($Y$) was calculated as:

$$Y = -234.42 + 17.72A + 1.62B + 3.67C - 0.02AB - 0.06AC + 0.03BC \\ - 0.86A^2 - 0.03B^2 - 0.03C^2.$$

The variance analysis of the RSM analysis model is shown in Table 8. The default term had a $p$-value of 0.0845, which exceeds 0.05, indicating the model's soundness. The determination coefficient $R^2$ and adjusted coefficient Adj $R^2$ were 0.9970 and 0.9932, respectively, demonstrating an excellent correlation between the theoretical and actual values of IPS content. The F-values of factors $A$, $B$, and $C$ were 581.67, 49.24, and 181.67, respectively, indicating the precedence of their effects on IPS content as $A > C > B$. Factors $A$, $B$, $C$, and the interaction terms $AC$, $BC$, $A^2$, $B^2$, and $C^2$ exhibited statistically significant effects on IPS content ($p < 0.05$).

### RSM analysis

The RSM analysis results demonstrated the interaction effects between yeast extract concentration, *S. vaninii* inoculum ratio, and liquid loading volume on IPS content (Fig. 4). The contour plot visually represents the transition from the blue region to the

**Table 8 ANOVA of the fitted quadratic polynomial model.**

| Source | Sum of squares | df | Mean square | F-value | p-value |
|---|---|---|---|---|---|
| Model | 2,495.17 | 9 | 277.24 | 260.9 | <0.0001** |
| A: Yeast extract concentration | 618.11 | 1 | 618.11 | 581.67 | <0.0001** |
| B: ST inoculum ratio | 52.32 | 1 | 52.32 | 49.24 | 0.0002** |
| C: Liquid loading volume | 193.05 | 1 | 193.05 | 181.67 | <0.0001** |
| AB | 1.33 | 1 | 1.33 | 1.25 | 0.3006 |
| AC | 71.2 | 1 | 71.2 | 67 | <0.0001 |
| BC | 180.61 | 1 | 180.61 | 169.96 | <0.0001 |
| $A^2$ | 249.45 | 1 | 249.45 | 234.74 | <0.0001 |
| $B^2$ | 41.82 | 1 | 41.82 | 39.36 | 0.0004 |
| $C^2$ | 990.06 | 1 | 990.06 | 931.7 | <0.0001 |
| Residual | 7.44 | 7 | 1.06 | | |
| Lack of fit | 5.8 | 3 | 1.93 | 4.7 | 0.0845 |
| Pure error | 1.64 | 4 | 0.41 | | |
| Cor total | 2,502.61 | 16 | | | |
| $R^2$ | | | | 0.9970 | |
| Adj $R^2$ | | | | 0.9932 | |

**Note:**
**$p < 0.01$.

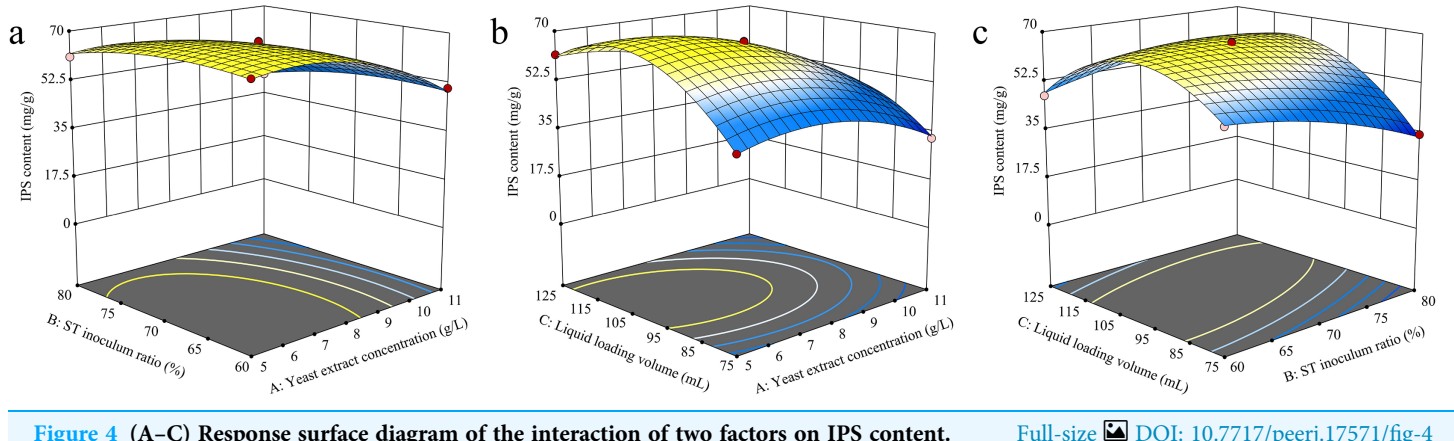

**Figure 4 (A–C) Response surface diagram of the interaction of two factors on IPS content.**

yellow area, with a steeper gradient indicating more significant results. The surfaces of yeast extract concentration, *S. vaninii* inoculum ratio, and liquid loading volume exhibited a high degree of curvature, suggesting a highly significant impact of these three factors on IPS content.

Design-Expert 13 software was used to optimize the cultivation conditions for IPS content, and the optimal culture conditions were determined to be a yeast extract concentration of 6.12 g/L, *S. vaninii* inoculum ratio of 68.87%, and a liquid loading volume of 105.55 mL. Under these conditions, the projected IPS content was calculated to be 69.4559 mg/g. However, due to practical constraints, the yeast extract concentration was

**Table 9 Comparison of biomass and IPS content before and after optimisation of co-culture culture conditions.**

| Indicator | Unoptimized | Optimized | Optimized to increase the percentage |
|---|---|---|---|
| Biomass (g/100 mL) | 0.7651 ± 0.0069 | 0.7628 ± 0.0381 | −0.3% |
| IPS content (mg/g) | 59.7789 ± 0.3635 | 69.9626 ± 2.1491 | 17.04% |

adjusted to 6 g/L, the *S. vaninii* inoculum ratio was recalibrated to 69%, and the liquid loading volume was adjusted to 106 mL.

### Regression model validation

The results obtained from the optimization design experiments of the RSM showed that the IPS content was 69.4559 mg/g under optimal cultivation conditions, which was 16.19% higher than the IPS content under the pre-optimized culture conditions. Furthermore, when the optimum conditions determined by the RSM optimization method were applied in the actual culture, the IPS content reached 69.9626 mg/g, with a deviation of only 0.73% from the projected value (Table 9). These findings highlight the accuracy and reliability of the projected results obtained through RSM optimization.

## Antioxidant capacity analysis

### Analysis of DPPH free radical scavenging capacity

The scavenging rate of DPPH radicals by *S. vaninii*, *P. sapidus*, and the *S. vaninii* and *P. sapidus* co-culture increased with the higher mass concentration of IPS. At a mass concentration of 0.02 mg/mL, the scavenging rates were 3.71%, 6.26%, and 7.20% for *S. vaninii*, *P. sapidus*, and the *S. vaninii* and *P. sapidus* co-culture, respectively. As the mass concentration increased to 0.4 mg/mL, the scavenging rates for *S. vaninii* and the *S. vaninii* and *P. sapidus* co-culture rapidly rose to 87.10% and 80.23%, stabilizing around 87% and 81%, respectively. When the mass concentration reached 0.6 mg/mL, the *P. sapidus* scavenging rate peaked at 84.99% and remained stable at around 85% (Fig. 5A). The mass concentration of antioxidants ($IC_{50}$) at a scavenging rate of 50% was used as the evaluation criterion to assess the antioxidant performance and free radical scavenging ability of antioxidants. A lower $IC_{50}$ value indicates a higher ability of antioxidants to scavenge free radicals (*Tang et al., 2010*). The $IC_{50}$ values of IPS from *S. vaninii*, *P. sapidus*, and the *S. vaninii* and *P. sapidus* co-culture were calculated as 0.1875, 0.2581, and 0.1687 mg/mL, respectively. This suggests that IPS from *S. vaninii* and *P. sapidus* co-cultures exhibit a stronger ability to scavenge DPPH free radicals compared to pure cultures of *S. vaninii* and *P. sapidus*.

### Analysis of ABTS free radical scavenging capacity

The scavenging rate of ABTS free radicals by *S. vaninii*, *P. sapidus*, and the *S. vaninii* and *P. sapidus* co-culture increased as the mass concentration of IPS increased. At a mass concentration of 0.02 mg/mL, the scavenging rates were 25.62%, 24.86%, and 25.19% for *S. vaninii*, *P. sapidus*, and the *S. vaninii* and *P. sapidus* co-culture, respectively. As the mass

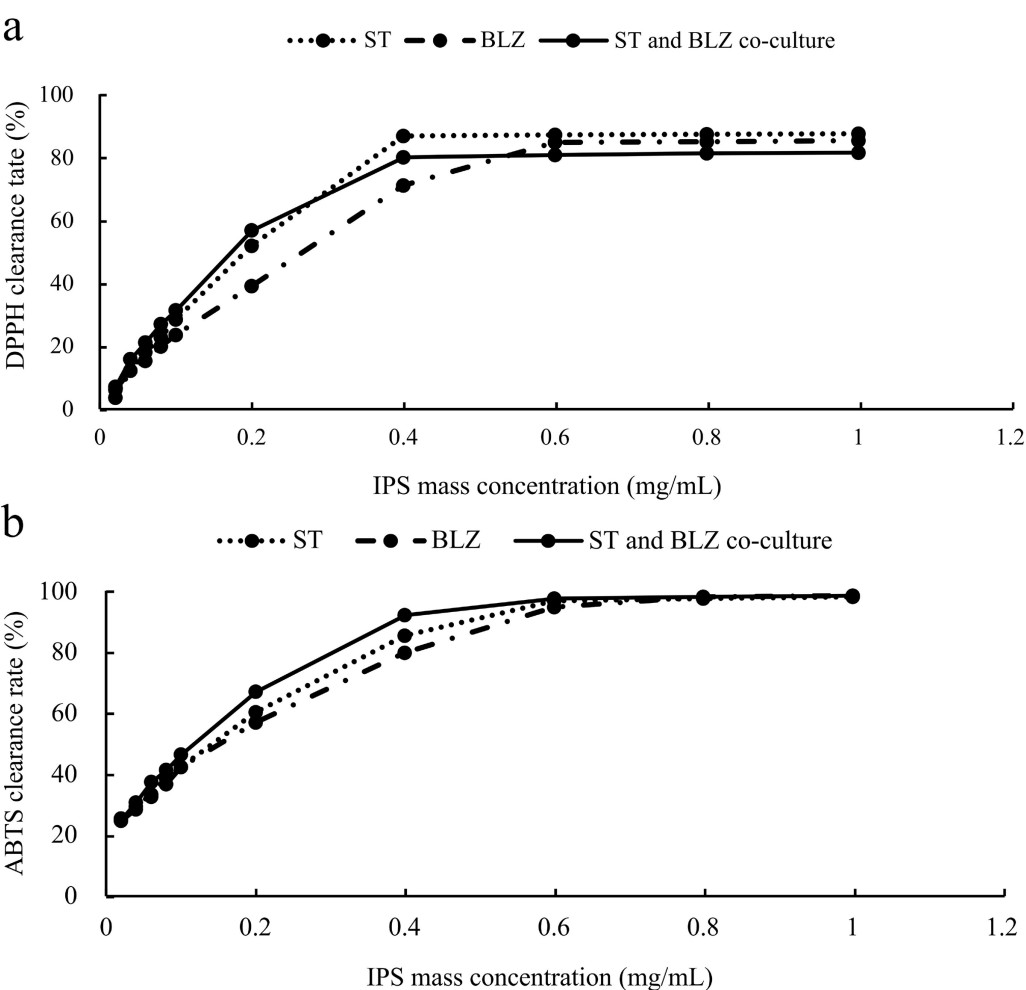

**Figure 5 Antioxidant capacity of IPS samples.** (A) Ability to scavenge DPPH radicals; (B) ability to scavenge ABTS radicals.

concentration of IPS increased to 0.6 mg/mL, the scavenging rates rapidly rose to 97%, 95.05%, and 97.86% for *S. vaninii*, *P. sapidus*, and the *S. vaninii* and *P. sapidus* co-culture. With further increases in mass concentration, the clearance remained around 98% (Fig. 5B). The IC$_{50}$ of IPS on ABTS radical scavenging for *S. vaninii*, *P. sapidus*, and the *S. vaninii* and *P. sapidus* co-culture were calculated to be 0.1397, 0.1515, and 0.1136 mg/mL, respectively. This indicates that the IPS of *S. vaninii* and *P. sapidus* co-cultures exhibit a stronger scavenging ability of ABTS radicals compared to the IPS of *S. vaninii* and *P. sapidus* pure cultures.

## DISCUSSION

This study aimed to investigate the impact of co-culture conditions on IPS content by creating an artificial symbiosis platform to mimic natural fungal interactions. Initially, liquid co-cultures were conducted with seven different fungi and *S. vaninii*. Results indicated that the co-culture of *S. vaninii* and *P. sapidus* exhibited the most significant increase in IPS content, reaching 59.8302 mg/g. It is important to note that not all fungal

co-cultures led to an increase in polysaccharide content, highlighting the importance of considering strain interaction, culture medium selection, and growth rate. Other researchers have also explored co-cultures of various fungi (*Peng et al., 2023*). For instance, *Yao et al. (2016)* established 136 fungus-fungus symbiosis systems by co-culturing 17 basidiomycete species, identifying optimal interactions. The co-culture of *Trametes versicolor* and *Ganoderma applanatum* demonstrated the most favorable outcome. To achieve an effective co-culture of *S. vaninii* and *P. sapidus*, different co-cultivation combinations were tested, with combination 8 proving most effective. This combination involved pre-culturing *S. vaninii* and *P. sapidus* seeds separately for 2 days and 0 days, respectively, followed by liquid co-culturing. *Guo et al. (2021)* conducted experiments on six co-culture models of *Sanghuangporus lonicericola* and *Cordyceps militaris*, determining that pre-fermentation of *S. lonicericola* and *C. militaris* for 3 days and 1 day, respectively, before co-fermentation was optimal. These findings emphasize the variability in co-culture effects among different fungi, highlighting the importance of selecting appropriate culture combinations based on specific production requirements. As shown in Fig. 2 of this study, selecting combination 1 for subsequent experiments is recommended when considering only the biomass of mycelial balls. Fungi, as a significant source of natural pigments (*Meruvu & dos Santos, 2021*), have garnered attention for their versatile applications in various industries including food and beverage, textile, and medical fields (*Abel et al., 2023*). The 12 culture combinations depicted in Fig. 2 exhibit significant differences in the color of the fermentation broth, suggesting potential research directions for true pigments.

Medium components and culture conditions play a crucial role in the growth of microorganisms and in the production of secondary metabolites (*Sun, 2022*). Therefore, it is essential to optimize both the medium components for microbial growth and the culture conditions necessary to produce secondary metabolites. RSM can elucidate the relationship between different variables and is an effective approach for optimizing culture medium components and co-culture conditions. In this study, single-factor tests, PB tests, and RSM showed that IPS content is significantly influenced by yeast extract powder concentration, liquid filling volume, and *S. vaninii* inoculation volume proportion. When using optimal co-culture conditions, the IPS content was 69.9626 mg/g, with a deviation of only 0.73% from the predicted value of 69.4559 mg/g. Thus, the findings of this study suggest that RSM is suitable for optimizing the co-culture conditions of *S. vaninii* and *P. sapidus*.

*Sanghuangporus vaninii* and *P. sapidus* are two fungi known for their significant medicinal properties. They have been found to enhance the body's immune system, regulate blood lipids, exhibit antibacterial and anti-tumor effects, and produce active polysaccharides (*Li, Bau & Yang, 2022*; *Yang et al., 2023*). Free radicals, constantly generated and removed in the body, can lead to cell damage, accelerated aging, and various diseases, including cancer, if accumulated in excess (*Wang et al., 2014*). Therefore, the antioxidant and free radical scavenging properties of fungal polysaccharides play a crucial role in maintaining health. Results from this study indicate that the $IC_{50}$ values for IPS co-cultured with *S. vaninii*, *P. sapidus*, and the *S. vaninii* and *P. sapidus* co-culture on DPPH and ABTS free radical scavenging rates show stronger antioxidant capacity when

co-cultured with both *S. vaninii* and *P. sapidus* compared to single cultures with either *S. vaninii* or *P. sapidus*. Antioxidants currently in use are typically synthetic, such as butylhydroxyanisole and 2,6-bis(1,1-dimethylethyl)-4-methylphenol. While these antioxidants can enhance product stability and maintain quality, they also come with various drawbacks and potential risks to human health (*Cai et al., 2019*). Thus, the search for new and safe natural antioxidants is of utmost importance. The conventional pure culture method has limitations in enhancing the antioxidant activity of fungal polysaccharides. However, the co-culture method employed in this study resulted in fungal polysaccharides with stronger antioxidant capabilities compared to those from single cultures of each strain, suggesting a promising avenue for the development and utilization of antioxidants.

## CONCLUSIONS

This study screened *P. sapidus* strains through liquid co-culturing to identify those suitable for co-culturing with *S. vaninii* and those capable of increasing IPS content. After evaluating 10 co-culture combinations of *S. vaninii* and *P. sapidus*, differences were observed in terms of IPS content, biomass, color of the fermentation broth, and the size, shape, and color of the bacterial balls. Combination 8 demonstrated the most favorable outcomes, where the seed pre-culture solutions of *S. vaninii* and *P. sapidus* were initially cultured for 2 days and 0 days, respectively, before co-culturing. Through single factor experiments, PB, and RSM, the co-culture conditions of *S. vaninii* and *P. sapidus* were optimized to enhance IPS production in a shorter timeframe. The IPS content achieved under the optimized conditions was 69.9626 mg/g, representing a 447.16% increase compared to the single culture of *S. vaninii* (12.7866 mg/g) and a 34.69% increase compared to the single culture of *P. sapidus* (51.9439 mg/g). These results indicate that the method proposed in this study effectively boosts IPS content in co-cultured *S. vaninii* and *P. sapidus*, showcasing promising feasibility. The analysis of antioxidant activity indicated that IPS co-cultured with multiple strains exhibited stronger scavenging abilities against DPPH and ABTS free radicals compared to polysaccharides cultured with a single strain. These findings offer valuable insights for the selection of fungal co-culture strains and combinations, as well as lay a foundation for the future development and utilization of fungal polysaccharides. However, this is a preliminary study, and investigating potential changes in product components or generating new substances will require further research using metabolomics. Additionally, understanding the molecular mechanisms of co-culturing through transcriptomics is essential for advancing this area of research.

### Funding

This work was supported by the National Key Research and Development Program of China (No.2017YFD0300104). The funders had no role in study design, data collection and analysis, decision to publish, or preparation of the manuscript.

## Grant Disclosures

The following grant information was disclosed by the authors:
National Key Research and Development Program of China: 2017YFD0300104.

## Competing Interests

The authors declare that they have no competing interests.

## Author Contributions

- Yuantian Lu conceived and designed the experiments, performed the experiments, analyzed the data, prepared figures and/or tables, authored or reviewed drafts of the article, and approved the final draft.
- Di Liu conceived and designed the experiments, analyzed the data, authored or reviewed drafts of the article, and approved the final draft.

## Data Availability

   The raw measurements are available in the Supplemental Files.

## Supplemental Information

Supplemental information for this article can be found online at http://dx.doi.org/10.7717/peerj.17571#supplemental-information.

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
