# Peer review of "Optimization of polysaccharide conditions and analysis of antioxidant capacity in the co-culture of Sanghuangporus vaninii and Pleurotus sapidus"

_PeerJ, doi:10.7717/peerj.17571_

## Round 0.1 · original submission · Major Revisions

- Please clearly state the methodology. There are several experiments that appear unclear and are not directly linked to the results, and subsequently, to the conclusion.

- Additional experiments are necessary to identify the specific types of polysaccharides produced.

**Language Note:** The review process has identified that the English language must be improved. PeerJ can provide language editing services - please contact us at [email protected] for pricing (be sure to provide your manuscript number and title). Alternatively, you should make your own arrangements to improve the language quality and provide details in your response letter. – PeerJ Staff

Reviewer 1 ·

Basic reporting

1. English writing should be improved
Line 31: conditions -> conditions
Line 33: demonstrate -> demonstrated
Line 77: Pleurotus sapidus -> P. sapidus
Line 214: strain -> strains
Line 314: are -> were
Line 334: is -> was
Line 348: exhibit -> exhibited
Line 358: indicate -> indicated
...
2. Line 503: Incorrect reference format
3. Figure 3-10: It will better if these images can be merged into one figure, accordingly, the description of the results should be highly condensed
4. ST and BLZ compete each other. Why do competing strains can produce more fungal polysaccharides when co-cultured? Is there enough theoretical evidence? If so, I suggested that this be described in detail in the introduction, so that readers can easily understand why co-culture is a better approach than monoculture.

Experimental design

1. Line 337: "The aim was to discover new polysaccharide compounds or enhance the production of existing ones", the study did not identify any new polysaccharide compound, e.g. reddish-brown substances
2. In the study, liquid culture was applied, but not solid culture. I wonder that what are the effects of these two methods on yield of fungal polysaccharides
3. Why should biomass be measured and how does it relate to the main aims of the study?

Validity of the findings

1. Line 226-230: It would be better to specify that the Mode 11 and 12 indicated the independent growth of BLZ and ST, respectively
2. Line 416-418: "different fungi require different co-culture modes", co-culture modes are influenced by growth rate or other physiological features, or culture conditions, If the influencing factor is not clear or too complex, the so-called "different co-culture modes" will have little substantial guiding significance for future studies
3. Line 439: "447.16%", please check if the result is correct

·

Basic reporting

The manuscript discusses improving polysaccharide production by co-culturing Sanghuangporus vaninii and Pleurotus sapidus under optimized conditions. However, it has several errors and areas that need improvement. A thorough revision is recommended to fix these errors and improve the manuscript's quality. This will make the research clearer and more accurate, enhancing its value.


The manuscript contains numerous typographical errors and inaccuracies in its current form. To enhance the overall clarity and professional tone of the document, it is recommended that the authors undertake a thorough revision of the English language used throughout. Attention to detail in correcting these errors will significantly improve the readability and credibility of the manuscript. Such as:
Line 20: aimed
There should be a space before the opening parenthesis of the citation. (Line: 39, 45, 47, 62, 68, 71-73, 78, 81, 84, 372, 375, 381, 385, 391, 396, 403, 422, 425, 428)
Line 21: Remove S. vaninii and P. sapidus
Line 22: Please consider changing “with” to “using or employing”
Line 23: Adding "a" before "Plackett-Burman (PB) design" for grammatical consistency.
Line 24, 90: Removing "design" after "RSM optimization" to avoid redundancy, as "RSM" already implies a design or methodology.
Line 25: "seeds culture fluid" Please change to "seed cultures" for clarity and correctness.
Line 26: "followed by co-culture" Please change to "followed by co-culturing" to maintain tense consistency.
Line 27: "significantly higher polysaccharide content" Please change to "a significantly higher polysaccharide content compared to" to provide a clearer comparison with monocultural conditions.
Line 27-31: Please consider restructuring for clarity, breaking it into two sentences."Among these, yeast extract concentration was found to have the greatest influence, followed by liquid loading volume and the S. vaninii inoculum ratio".
Line 32: Please consider changing "representing" to "which represents" to improve the flow and clarity of the sentence.
Line 34: Slightly rephrase the last part to "enhance fungal polysaccharide production" for a more direct and fluid statement.
Line 38: Please remove "an"
Line 55: Please consider Adding "the" before "physical environment" for grammatical precision.
Line 62: Please remove "." after "metabolites"
Line 90: Please consider changing "will enhance" to "are expected to enhance".
Line 124: Please consider changing " 5 ST agar blocks and 1 BLZ agar block" to " 5 agar blocks of ST and 1 agar block of BLZ".
Line124: Please consider adding "aseptically" in front of transferred.
Line 208: Please consider changing "inoculated separately" to " separately inoculated".
Line 405-406: Please consider changing"Sanghuangporus lonicericola and Cordyceps militaris" to " S. lonicericola and C. militaris".

Experimental design

The authors utilized a Single-Factor Experiment, addressing each condition individually. However, it remains unclear whether the authors transitioned to the subsequent condition based on the optimal outcomes observed in the preceding tests. A clarification regarding the selection process for advancing to the next condition would enhance the understanding of the experimental design and decision-making rationale.

Validity of the findings

In the introduction, the authors articulated the primary objective of the study as investigating the interaction between two fungi through confrontation culture on both agar plates and broth media. However, in the subsequent discussion, the stated aim shifted towards the exploration of novel polysaccharide compounds or the augmentation of existing ones. Regrettably, the results section did not provide conclusive evidence regarding the discovery of new polysaccharide compounds. This discrepancy raises a noteworthy point of clarification. While the confrontation culture method was employed to discern fungal interactions, the outcomes were predominantly discussed in terms of biomass and polysaccharide content. The specific identification or characterization of novel polysaccharide compounds, as initially indicated as an objective, was not substantiated by the experimental findings. To enhance the rigor of the study and fulfill the stated objectives comprehensively, future investigations might consider incorporating analytical techniques capable of identifying and characterizing polysaccharide compounds explicitly. This refinement would align the study's outcomes more closely with the initially outlined aim of uncovering new polysaccharide compounds or enhancing the production of existing ones.

Additional comments

1. The authors are encouraged to delve more comprehensively into the underlying mechanisms responsible for the observed enhancement in polysaccharide production resulting from the co-culture of the two fungi. A more thorough exploration of the intricacies of gene regulation, specifically detailing how the co-culture influences and modulates gene expression, would significantly enrich the discussion. Expanding the discourse to encompass the molecular and genetic aspects involved in the cooperative interaction between the fungi would provide a more nuanced understanding of the observed phenomena. This refinement is essential to elevate the depth and scholarly impact of the discussion section, ensuring a more robust exploration of the scientific underpinnings of polysaccharide production enhancement in the context of fungal co-culture.

2. The authors are encouraged to underscore the novel contributions and unique insights derived from the experimental endeavors. A specific emphasis on whether this study marks the inaugural attempt to co-culture Sanghuangporus vaninii and Pleurotus sapidus would be beneficial. By addressing this aspect, the authors can elucidate the novelty of their research within the broader scientific context. Additionally, a more explicit delineation of the distinctive findings and knowledge generated through this co-culturing endeavor would fortify the manuscript's scholarly impact. Clearly articulating the advancements or departures from existing knowledge will reinforce the significance of the study and contribute to the overarching scientific discourse on the co-cultivation of these fungal species.

Reviewer 3 ·

Basic reporting

The authors write clearly and objectively. In the PDF I indicated some points to clarify or explain. The citations and references must be carefully revised.

Experimental design

The authors present a rigorous investigation. However, they should explain why no citation is made in the Materials and Methods, as well as clarify what strains were used.

Validity of the findings

no comment

Additional comments

no comment

Annotated reviews are not available for download in order to protect the identity of reviewers who chose to remain anonymous.

---

## Round 0.2 · Minor Revisions

Please carefully address the reviewers' remaining comments.

Reviewer 1 ·

Basic reporting

no comment

Experimental design

no comment

Validity of the findings

Question 2:
My expression may have misled you, and what I mean is that the statements like "different co-culture modes" are not clear or specific.

Additional comments

no comment

·

Basic reporting

The author has improved the writing.

Experimental design

No comment

Validity of the findings

please see additional comment

Additional comments

1. In the seed culture fluid, the author mentioned bacterial blocks. It should be fungal mycelial blocks instead.

2. Table 4, please provide statistical tests.

3. From Figure 2 and Table 1, does “0 day” mean no need for pre-culture liquid culture?

4. Why choose model 8 when model 1 does not require pre-culturing

5. From line 262-263, why model 8 is more prominent than model 1?

6. When authors mentioned inoculation volume, does it mean inoculation of 2 strain in total?
For validation, could you please show the table or figure comparing the unoptimized and optimized conditions

Reviewer 3 ·

Basic reporting

no comment

Experimental design

The main questions that remain are (i) the information on used strains is still unclear. As commented in the PDF, the strains must be trackable and available to use in other studies, and (ii) volume information in the Liquid co-culture methods section.

Validity of the findings

no comment

Additional comments

The study was improved. The study is methodically supported and offers direct and indirect results, with possible long-term methods for future studies. The methods and species to co-culture are useful for obtaining fungal polysaccharides.

Please, make sure to consult the PDF commentaries and direct suggestions.

Annotated reviews are not available for download in order to protect the identity of reviewers who chose to remain anonymous.

---

## Round 0.3 · accepted · Accept

The revised version of the manuscript has improved significantly and can be accepted for publication.